# Leveraging supervised learning for functionally informed fine-mapping of cis-eQTLs identifies an additional 20,913 putative causal eQTLs

Qingbo S. Wang [1,2,3 ✉], David R. Kelley [4], Jacob Ulirsch [1,2,5], Masahiro Kanai [1,2,3,6], Shuvom Sadhuka[1,7], Ran Cui[1,2], Carlos Albors[1,2], Nathan Cheng[1,2], Yukinori Okada [6,8,9], The Biobank Japan Project*, Francois Aguet [1], Kristin G. Ardlie[1], Daniel G. MacArthur [10,11] & Hilary K. Finucane [1,2 ✉]

The large majority of variants identified by GWAS are non-coding, motivating detailed characterization of the function of non-coding variants. Experimental methods to assess variants' effect on gene expressions in native chromatin context via direct perturbation are low-throughput. Existing high-throughput computational predictors thus have lacked large gold standard sets of regulatory variants for training and validation. Here, we leverage a set of 14,807 putative causal eQTLs in humans obtained through statistical fine-mapping, and we use 6121 features to directly train a predictor of whether a variant modifies nearby gene expression. We call the resulting prediction the expression modifier score (EMS). We validate EMS by comparing its ability to prioritize functional variants with other major scores. We then use EMS as a prior for statistical fine-mapping of eQTLs to identify an additional 20,913 putatively causal eQTLs, and we incorporate EMS into co-localization analysis to identify 310 additional candidate genes across UK Biobank phenotypes.

[1] Broad Institute of MIT and Harvard, Cambridge, MA, USA. [2] Analytic and Translational Genetics Unit, Massachusetts General Hospital, Boston, MA, USA. [3] PhD program in Bioinformatics and Integrative Genomics, Harvard Medical School, Boston, MA, USA. [4] Calico Life Sciences, South San Francisco, CA, USA. [5] PhD program in Biological and Biomedical Sciences, Harvard Medical School, Boston, MA, USA. [6] Department of Statistical Genetics, Osaka University Graduate School of Medicine, Osaka, Japan. [7] Harvard College, Cambridge, MA, USA. [8] Laboratory of Statistical Immunology, Immunology Frontier Research Center (WPI-IFReC), Osaka University, Osaka, Japan. [9] Integrated Frontier Research for Medical Science Division, Institute for Open and Transdisciplinary Research Initiatives, Osaka University, Osaka, Japan. [10] Centre for Population Genomics, Garvan Institute of Medical Research, Darlinghurst, NSW, Australia. [11] Centre for Population Genomics, Murdoch Children's Research Institute, Parkville, VIC, Australia. *A list of authors and their affiliations appears at the end of the paper. ✉email: qingbow@broadinstitute.org; finucane@broadinstitute.org

Although genome-wide association studies (GWAS) have identified large numbers of loci associated with complex traits[1,2], identifying the underlying biological mechanisms is often difficult. Two particular challenges are that (1) the majority of the associated variants are in noncoding regions[1], and (2) the association signals from GWAS studies typically contain a large number of variants in linkage disequilibrium (LD)[3]. Interpreting associations in GWAS to identify the underlying causal mechanisms requires an understanding of the function of noncoding variants at single-variant resolution.

Many approaches to characterize noncoding variants exist. Large-scale consortium studies[4,5] have provided a map of functional and regulatory elements across the genome in different cell types that are enriched in various trait heritability[6–10]. Reporter assays have been powerful tools to test variant effects in cellular contexts, but typical high-throughput massive parallel reporter assays (MPRAs)[11,12] do not represent the native chromatin context in the human genome. Direct introduction of single base pair variants in the native genome are still low throughput[13]. RNA-seq studies combined with genotyping or whole-genome sequencing have highlighted loci that are associated with gene expression in humans (eQTLs)[14–16]. However, as with GWAS, eQTL studies associate loci, rather than individual causal variants, to gene expression.

Statistical fine-mapping[3,17,18] is used to disentangle tightly correlated structures of the nearby genetic variants in LD to elucidate causal variant(s) in a locus identified by a genetic association study, such as a GWAS on an eQTL study. For example, Benner et al.[19] uses stochastic search to enumerate and evaluate possible causal configurations, and Wang et al.[20] performs iterative Bayesian stepwise selection to prioritize causal variants. Such fine-mapping methods have been applied to identify putative causal eQTLs (i.e., variants that modify gene expression in native chromatin context) that are valuable both for understanding gene regulation and for interpreting GWAS signals at a locus[15,16,21–24]. However, fine-mapped eQTLs fall short of genome-wide characterization of noncoding function, as many variants fail to be identified because of LD or small effect size.

While not providing the same level of confidence as genome editing or fine-mapped eQTLs, computational predictions are informative about variant function in native chromatin in human cells, and can be applied to every variant in the genome. For example, state-of-the-art computational methods predict the effects of noncoding genetic variants on the epigenetic landscape and on gene expression as a function of sequence context, using deep neural networks[25–30]. These methods, rather than directly training on gold standard expression-modifying variants, instead predict expression level or other outcomes as a function of sequence, and then score variants based on the difference in predicted expression between the two alleles.

Here, we combine such computational predictions with the large-scale, though not comprehensive, gold standard data provided by statistical fine-mapping of eQTLs, with two goals: to improve on existing computational predictors, and to expand the set of confidently identified eQTLs. Toward the former goal, we combine an existing sequence-based predictor[28] with epigenetic data and other gene features into a single predictor, leveraging fine-mapped eQTLs (https://www.finucanelab.org/data) as training data. Specifically, we directly train a predictor of whether a variant modifies expression using 14,807 putative expression-modifying variant–gene pairs in humans as training data and utilizing 6121 features; we call the resulting prediction the expression modifier score (EMS). Toward the second goal, we use EMS as a prior for statistical fine-mapping of eQTLs (analogous to recently performed functionally informed fine-mapping of complex traits[31–33]), increasing fine-mapping resolution and

identifying an additional 20,913 variants across 49 tissues. Finally, using UK Biobank (UKBB)[34] phenotypes as an example, we show that EMS can be incorporated into colocalization analysis at scale, and we identify 310 additional candidate genes for UKBB phenotypes.

## Results

**Functional enrichment of fine-mapped eQTLs.** To define the set of putative expression-modifying variant–gene pairs, we analyzed results of recent fine-mapping of *cis*-eQTLs (±1 Mb window) from GTEx v8 (ref. [16]; https://www.finucanelab.org/data), including the 14,807 variant–gene pairs with posterior inclusion probability (PIP) > 0.9 according to two methods[19,20] across 49 tissues (Supplementary Figs. 1 and 2). The size of our dataset allowed us to quantify the enrichment of putative causal variant–gene pairs for several functional annotations, including deep learning-derived variant effect scores from Basenji[28,29] and distance to canonical transcription starting site (TSS), with high precision (Fig. 1, and Supplementary Figs. 3 and 4). Our results are consistent with previous studies[24,35]: putative causal variant–gene pairs are enriched for a number of functional annotations, such as 5'UTR, H3K4me3 (>10× enrichment compared to random variant–gene pairs) or distance to TSS (>500× enrichment for variant–gene pairs with distance to TSS < 100), but are not strongly enriched for introns (0.966×), and are depleted for a histone mark related to heterochromatin state (H3K9me3; 0.510× enrichment).

**Building a predictor for putative causal eQTLs [EMS].** Next, we built a random forest classifier of whether a given variant is a putative causal eQTL for a given gene using 807 binary functional annotations, including cell-type-specific histone modifications, as well as non-cell-type-specific annotations from the baseline model[4–6], 5313 Basenji features corresponding to functional activity predictors[28,29], and distance to TSS. We then scaled the output score of the random forest classifier to reflect the probability of observing a positively labeled sample in a random draw from all the variant–gene pairs (Fig. 2a and "Methods"), and named this scaled score the EMS. We performed the above process for 49 tissues in GTEx v8 individually, to obtain the EMS for variant–gene pairs in each tissue. In other words, EMS is an estimated probability of a variant–gene pair being a putative causal eQTL in a specific tissue, given the >6000 functional annotations of the variant–gene pair. For whole blood, the Basenji scores together had 55.0% of the feature importance for EMS, and distance to TSS had feature importance of 43.1%. The binary functional annotations together had <2% of importance (Fig. 2b, c). Analyses of other tissues also showed that (1) distance to TSS is by far the most important single feature, (2) Basenji scores individually explain a small fraction of predictor performance, but are collectively equally or more important than the distance to TSS, and (3) compared to the distance to TSS and Basenji scores, the feature importances of both cell-type-specific and nonspecific binary functional annotations are much smaller (Supplementary Data 1).

**Performance evaluation of EMS.** To evaluate the performance of EMS, we focused on whole blood and compared EMS (calculated by leaving one chromosome out at a time to avoid overfitting) to other genomic scores[26,36–39]. EMS achieved higher prediction accuracy than other genomic scores for putative causal eQTLs (top bin enrichment for held-out putative causal eQTLs 18.3× vs 15.1× for distance to TSS, the second best, Fisher's exact test $p = 3.33 \times 10^{-4}$, Fig. 3a; AUPRC = 0.884 vs 0.856 when using distance to TSS, the second best, Supplementary Fig. 5 and "Methods"). EMS

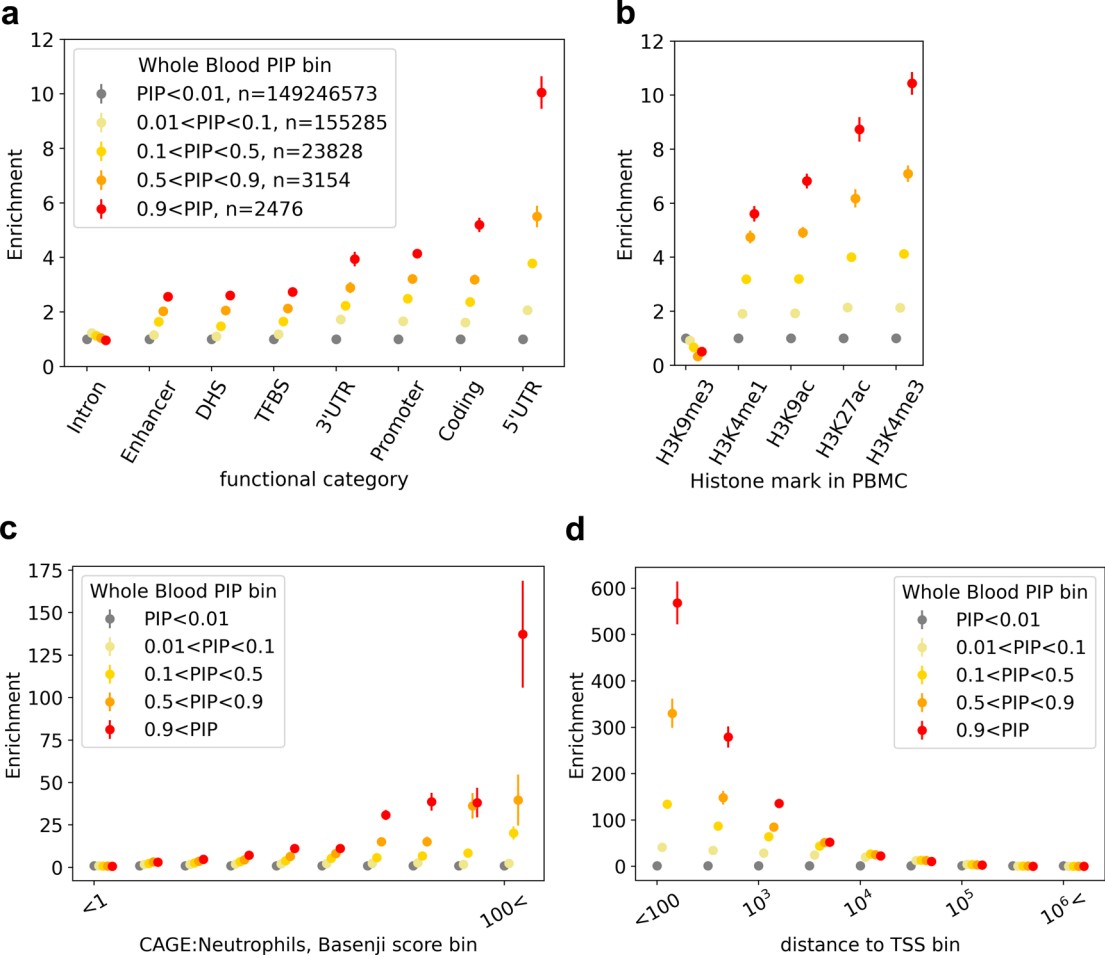

**Fig. 1 Examples of the enrichment of variant–gene pairs in whole-blood eQTL PIP bins for functional genomics features.** Enrichments of variant–gene pairs in different posterior inclusion probability (PIP) bins in binary functional features (non-tissue specific (**a**), tissue-specific in peripheral blood mononuclear cells (**b**), deep learning-derived regulatory activity (CAGE[46]) prediction in neutrophils (**c**), and distance to TSS (**d**) are shown (*n* is the number of variant–gene pairs).

was among the top-performing methods in prioritizing experimentally suggested regulatory variants from reporter assay experiments[12,40], despite not varying distance to TSS, the most informative feature (Fig. 3b, c, Supplementary Fig. 6, and "Methods"). Finally, EMS was also among the top-performing methods in prioritizing putative causal noncoding variants for hematopoietic traits in the UKBB dataset (17.6× for EMS, best, vs 17.1× for DeepSEA, the second best; Fig. 3d), although there are known differences between the genetic architectures of *cis*-gene expression and complex traits[41]. These results were consistent when we performed the same set of analyses in different datasets: hematopoietic traits in BioBank Japan[42] and lymphoblastoid cell line (LCL) eQTL in Geuvadis[14,22] (Supplementary Fig. 7).

**Functionally informed fine-mapping using EMS.** Since EMS is in units of estimated probability, one natural way to utilize EMS for better prioritization of putative causal eQTLs is to use it as a prior for statistical fine-mapping. We developed a simple algorithm for approximate functionally informed fine-mapping and applied it with EMS as a prior to obtain a functionally informed posterior, denoted $PIP_{EMS}$, in whole blood ("Methods"). As expected, we found that $PIP_{EMS}$ identified more putative causal eQTLs than the original PIP calculated with a uniform prior, denoted $PIP_{unif}$. Specifically, 95.4% of variants with $PIP_{unif} > 0.9$

also had $PIP_{EMS} > 0.9$ (2152 out of 2255), while only 33.8% of variants with $PIP_{EMS} > 0.9$ had $PIP_{unif} > 0.9$ (1125 out of 3277; Fig. 4a). Similarly, credible sets mostly decreased in size (Fig. 4b and Supplementary Data 2). Previous work in functionally informed fine-mapping[33] adjusted the prior so that the maximum prior value did not exceed 100 times the minimum prior value. We conducted a second round of functionally informed fine-mapping with a similar adjustment of the prior, identifying fewer additional putative causal eQTLs, as expected (1125 with EMS as a prior vs 269 with EMS adjusted to a max/min ratio of 100 as a prior; Supplementary Fig. 8).

We evaluated the quality of $PIP_{EMS}$ by comparing it with $PIP_{unif}$ and a publicly available eQTL fine-mapping result that uses distance to TSS as a prior[16,23] (denoted $PIP_{DAP-G}$) in two ways (other methods for functionally informed fine-mapping based on expectation maximization[31,32,35] would be computationally intensive for a dataset this size, while the recently introduced PolyFun[33] is designed for complex traits). First, $PIP_{EMS}$ had the highest enrichment level of reporter assay QTLs[40] (raQTLs) in the PIP > 0.9 bin (16.8× vs 12.9× in $PIP_{unif}$ and 11.4x in $PIP_{DAP-G}$, Fisher's exact test $p = 1.65 \times 10^{-2}$ between $PIP_{EMS}$ and $PIP_{DAP-G}$; Fig. 4c). Second, complex trait causal noncoding variants were comparably enriched in PIP > 0.9 bins (Supplementary Fig. 9). These results suggest that $PIP_{EMS}$ is a valid measure for identifying putative causal *cis*-regulatory variants.

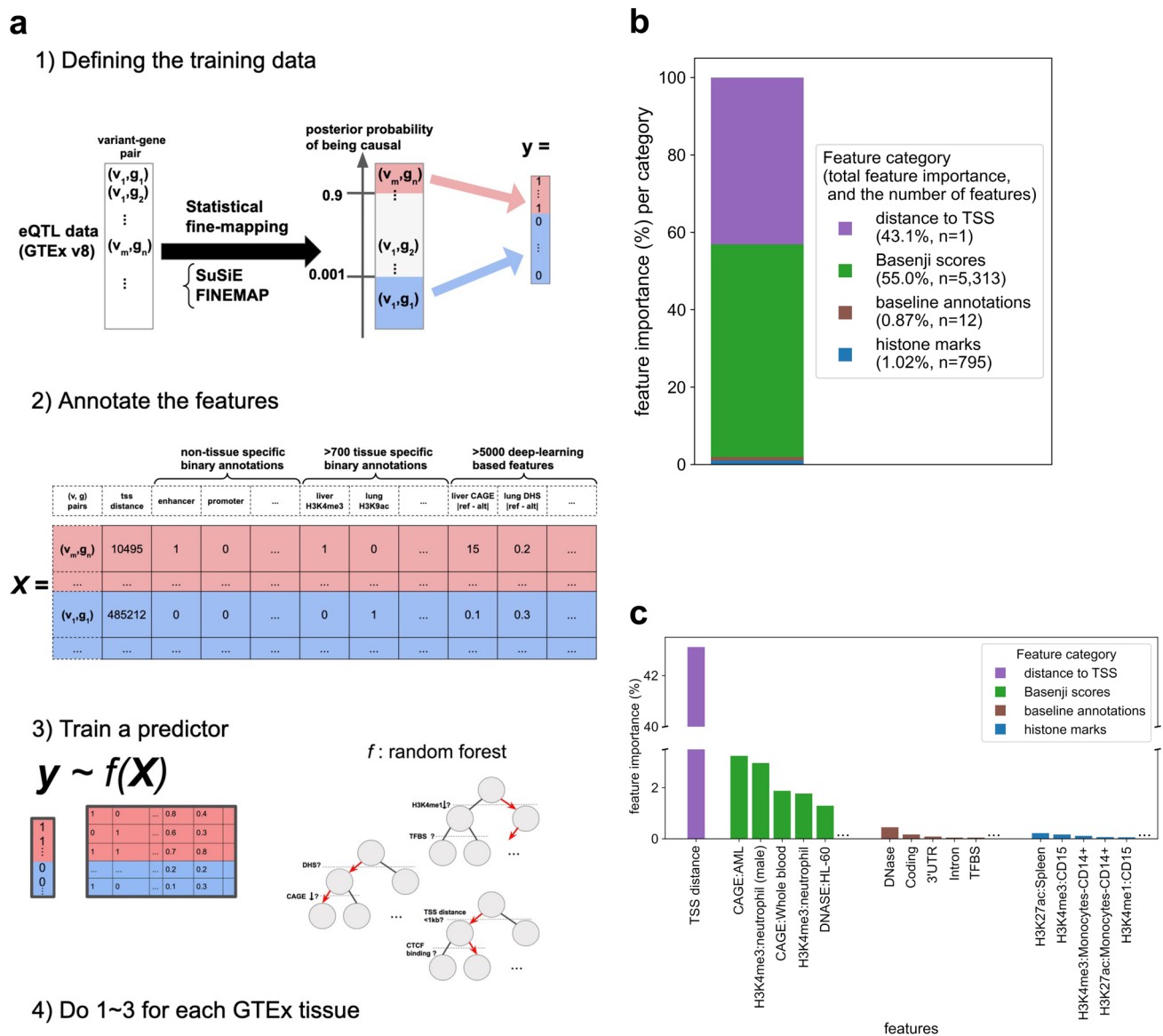

**Fig. 2 Schematic overview and feature importance of the expression modifier score (EMS). a** EMS is built by (1) defining the training data based on fine-mapping of GTEx v8 data, (2) annotating the variant–gene pairs with functional features, and (3) training a random forest classifier. We do this for each tissue. **b, c** Feature importance (mean decrease of impurity MDI[59]) for four different feature categories (**b**), and top features for each category (**c**). Baseline annotations are non-tissue-specific binary annotations from Finucane et al.[6], and histone marks are tissue-specific binary histone mark annotations from Roadmap[5]. In **b**, $n$ is the number of features in the category.

**Applying functionally informed PIP (PIP$_{EMS}$) in gene prioritization across 95 traits**. We next compared the utility of PIP$_{EMS}$ to PIP$_{unif}$ for complex trait gene prioritization, as in Weeks et al.[43]. To do this, we first calculated PIP$_{EMS}$ for 49 GTEx tissues using EMS of matched tissues as priors (Supplementary Figs. 10 and 11), resulting in a total of 20,913 additional eQTLs with PIP$_{EMS}$ > 0.9 (Fig. 5a, Supplementary Fig. 12, and Supplementary Data 3). Tissue-specificity of putative causal eQTLs were characterized by enrichments of corresponding tissue-specific transcription factor (TF) activity scores in the Basenji model (Fig. 5b–d, Supplementary Figs. 13 and 14, and "Methods"). We then colocalized the eQTL signals with 95 UKBB phenotypes. Using the evaluation gene set described in ref.[43], PIP$_{EMS}$ achieved higher precision and higher recall than PIP$_{unif}$ (Table 1 and "Methods"). Overall, PIP$_{EMS}$ elucidated 310 candidate genes for UKBB phenotypes that were not identified with PIP$_{unif}$

(Supplementary Data 4). On the other hand, PIP$_{DAP-G}$ showed lower precision than PIP$_{EMS}$ and PIP$_{unif}$ but higher recall (Table 1), suggesting the value of future studies in investigating different priors in eQTL fine-mapping and the trade-off between precision and recall for gene prioritization.

An example of PIP$_{EMS}$ resolving a credible set that is ambiguous with PIP$_{unif}$ is shown in Fig. 6. Here, four variants upstream of *CITED4* are in perfect LD in GTEx, giving PIP$_{unif}$ = 0.25 for all four (Supplementary Fig. 15). In UKBB, the four variants are also in high LD, with PIP for neutrophil count between 0.133 and 0.181 for all four. Thus, standard colocalization analysis does not identify *CITED4* as a neutrophil count-related gene (CLPP < $4.53 \times 10^{-2}$ for all variants; "Methods"). However, one of the four variants, rs35893233, creates a binding motif of *SPI1*, a TF known to be involved in myeloid differentiation[44,45], and presents epigenetic activity in myeloid-

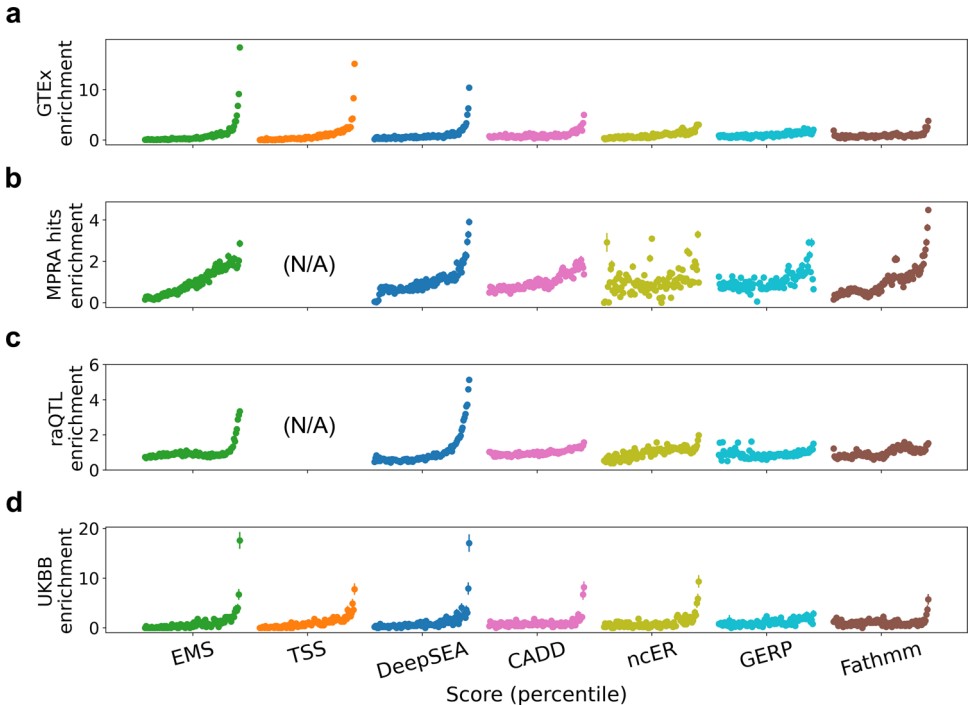

**Fig. 3 Performance evaluation of EMS.** Comparison of the different scoring methods in prioritizing putative causal whole-blood eQTLs in GTEx v8 (**a**), massive parallel reporter assay (MPRA) saturation mutagenesis hits[12] (**b**), reporter assay QTLs[40] (raQTLs) (**c**), and putative hematopoietic-trait causal variants in UKBB (**d**) in different score percentiles.

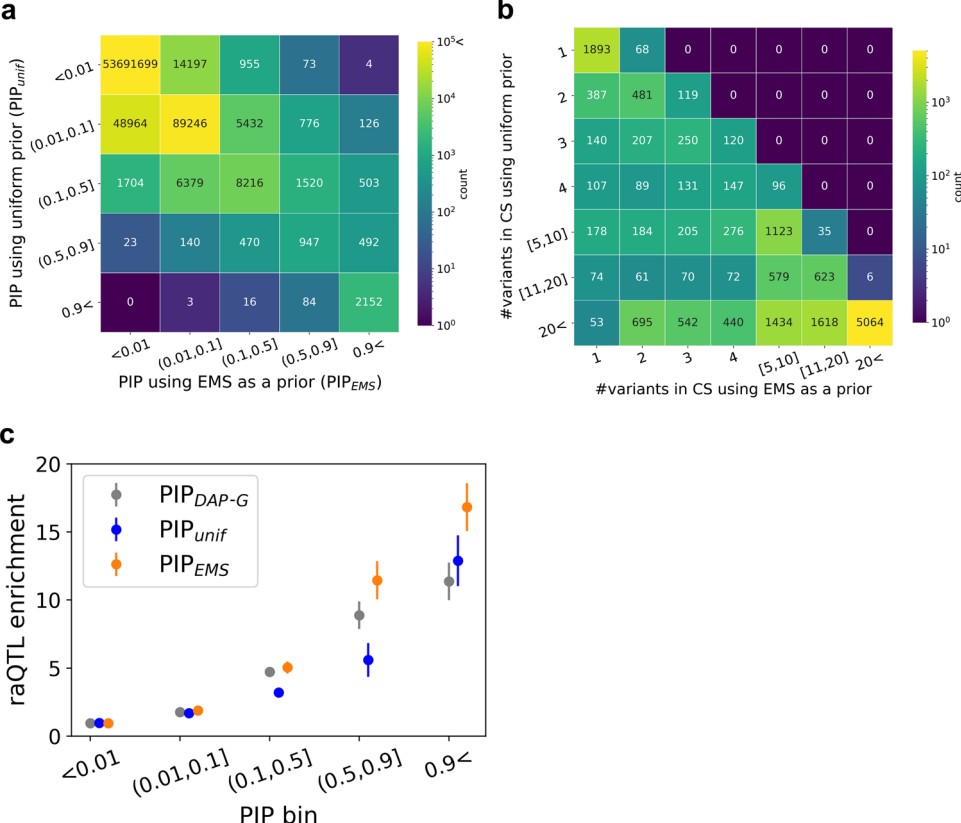

**Fig. 4 Functionally informed fine-mapping with EMS as a prior. a** Number of variant–gene pairs in different PIP bins using a uniform prior vs EMS as a prior. **b** Number of variants in the 95% credible set (CS) identified by fine-mapping with uniform prior vs EMS as a prior. **c** Enrichment of reporter assay QTLs (raQTLs) in different PIP bins (gray: publicly available eQTL PIP using DAP-G[23], blue: uniform prior, orange: EMS as a prior).

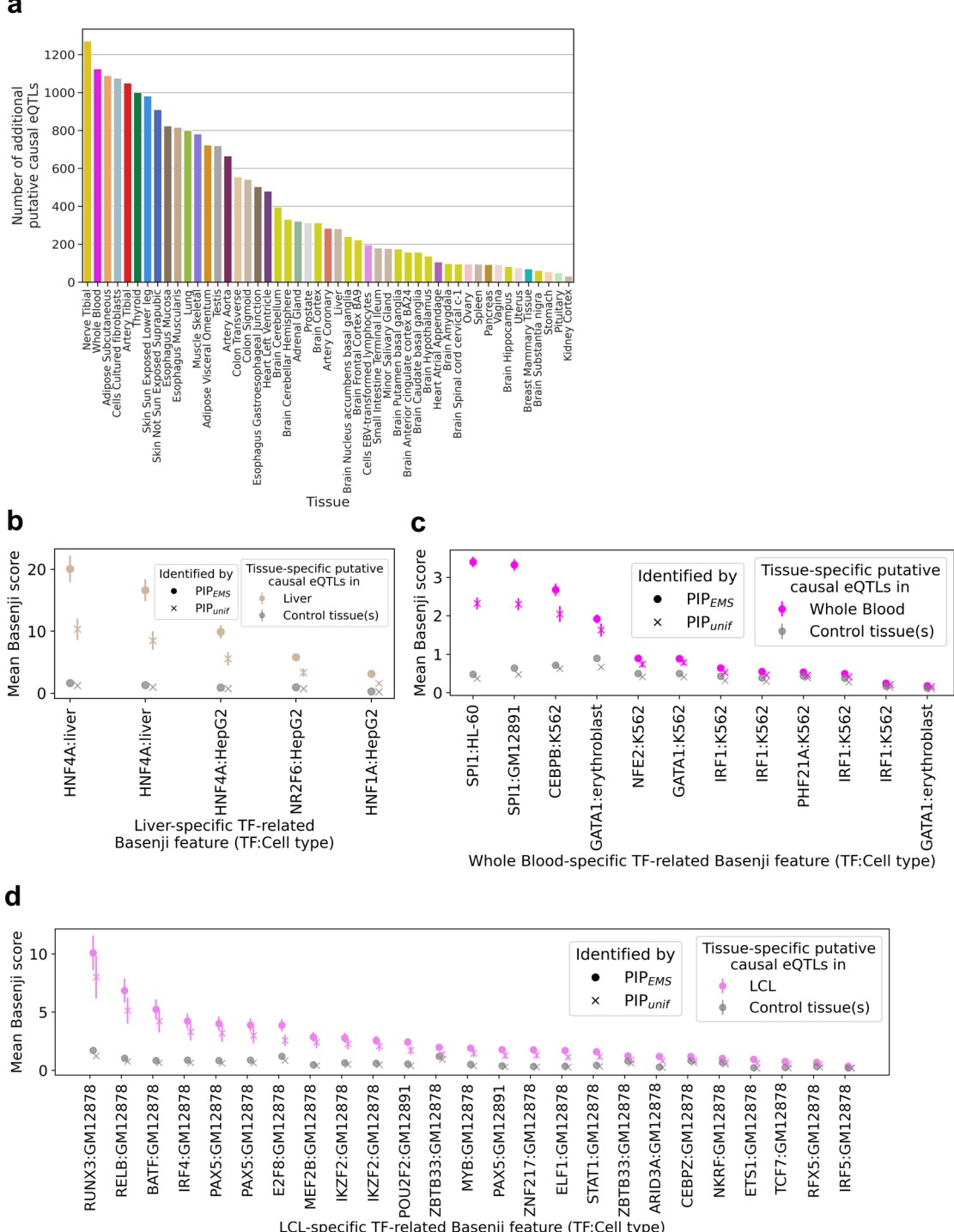

**Fig. 5 Functionally informed fine-mapping across 49 tissues. a** The number of additional putative causal eQTLs (defined by $PIP_{EMS} > 0.9$ and $PIP_{unif} < 0.9$) for each tissue is shown in descending order. **b–d** Mean Basenji score in different classes of tissue-specific putative causal eQTLs for tissue-specific TF-related Basenji features for liver (**b**), whole blood (**c**), and LCLs (**d**). In 39 out of all 42 features across all three tissues, the mean Basenji score in tissue-specific putative causal eQTLs identified by $PIP_{EMS}$ is significantly higher in the corresponding tissue than in control tissues (t test $p < 0.05/42$). This changes to 36 in 42 when using $PIP_{unif}$ instead of $PIP_{EMS}$. The enrichment of mean Basenji score in putative causal eQTLs in the corresponding tissue compared to control tissues is higher for $PIP_{EMS}$ than $PIP_{unif}$ for all 42 tissues ($p < 10^{-100}$ in aggregate), consistent with our understanding that functionally informed fine-mapping using EMS utilizes cell-type-specific functional enrichments, identified from putative causal eQTLs identified with a uniform prior, to identify additional putative causal eQTLs. Duplicated names are distinct features corresponding to biological replicates in the TF activity measurements. Out of 17,960 tissue-specific putative causal eQTLs, $n = 222$ were for liver (**b**), $n = 1758$ were for whole blood (**c**), and $n = 140$ were for LCL (**d**).

**Table 1 Precision and recall of the gene prioritization task for three different PIPs.**

| Method | Tool | Prior | Precision | Recall |
|--------|------|-------|-----------|--------|
| $PIP_{EMS}$ | SuSiE | EMS | 0.556 | 0.052 |
| $PIP_{unif}$ | SuSiE | Uniform | 0.525 | 0.039 |
| $PIP_{DAP-G}$ | DAP-G | Distance to TSS | 0.500 | 0.078 |

related cell types, such as showing the highest basenji score for cap analysis gene expression (CAGE)[46] activity in acute myeloid leukemia. This variant has >25× greater EMS than the other three variants ($1.73 \times 10^{-3}$ vs $6.11 \times 10^{-5}$, $1.00 \times 10^{-5}$ and $8.62 \times 10^{-6}$, respectively), enabling $PIP_{EMS}$ to narrow down the credible set to the single variant ($PIP_{EMS} = 0.956$ for rs35893233). Integrating EMS into the colocalization analysis thus allows identification of *CITED4* as a neutrophil count-related gene (CLPP = 0.173). Additional examples are described in Supplementary Fig. 16.

## Discussion

In this study, we introduced EMS, a prediction of the probability that a variant has a *cis*-regulatory effect on gene expression in a tissue. To derive EMS, we trained a random forest model that takes >6000 features. By analyzing the importance of each feature in the model, we showed that the importance of direct epigenetic measurements, such as binary histone mark peak annotation is relatively limited once distance to TSS and deep learning-derived variant effect scores (Basenji) were incorporated. Taking whole blood as an example, we showed that EMS accurately prioritizes putative causal eQTLs, reporter assay active variants, and putative complex trait causal noncoding variants. We provided a broader set of putative causal variants ($n = 20,913$ across 49 tissues) by using EMS as a prior to perform approximate functionally informed eQTL fine-mapping, and utilized EMS for colocalization analysis to identify 310 additional candidate genes for complex traits.

Evaluating predictors of noncoding variant function is complicated by the absence of gold standard data. While EMS outperformed other scores for prioritizing putative causal eQTLs, which we believe to be the closest to gold standard of existing large-scale base-pair resolution datasets, it did not outperform existing scores in prioritizing reporter assay active variants or putative complex trait causal noncoding variants. These latter two datasets, while valuable for independent validation, do not fully recapitulate the challenge of prioritizing causal expression-modifying variants in native context[41,47]. On the other hand, we recognize that putative causal eQTLs on a held-out chromosome do not constitute a fully independent validation set. As genome editing technologies continue to improve, we look forward to future large-scale datasets that will enable independent, gold standard evaluation and comparison of scores of noncoding functions at base-pair resolution.

Although our work refines our understanding of *cis*-gene regulatory mechanisms at single-variant resolution, it also presents limitations. First, there are biases in the way the training variants are ascertained: the power to call a putative causal variant is affected by the recombination rate and the allele frequency of the variant[48,49], and the GTEx cohort is highly biased towards adult samples with European ancestry background. Second, although we utilize over 6000 features in EMS, larger sets of variant and gene annotations, such as 3D configuration of genome[50,51], constraint[52–54], or pathway enrichment[43] of genes could allow us to further improve prediction accuracy. Third, we simplified the prediction task by thresholding PIP. We formed a

binary classification problem rather than a regression problem to build a predictor due to a highly skewed distribution of PIP, and because of LD-induced biases in variants with intermediate PIPs, but with larger sample size and a more principled hierarchical model, we could potentially take advantage of variants with intermediate PIP as well.

In this work, we focused on the task of predicting putative causal eQTLs. Future work could use a similar framework to predict putative causal splicing QTLs or other molecular QTLs for which statistical fine-mapping has identified a large number of high-PIP variants. In addition, although noisy effect size estimates from eQTL studies present a challenge, future work could explore leveraging features correlated with the sign and magnitude of effect (Supplementary Fig. 17) to estimate these values. As recent studies have suggested, such approaches would also be valuable in understanding the gene expression and complex trait regulation landscape in light of natural selection[55]. Our approach of utilizing statistical fine-mapping of eQTLs to define training data, assembling large number of features to train a predictor, and using the predictor output to expand the set of putative causal eQTLs is highly generalizable. EMS for all variant–gene pairs in GTEx v8 are publicly available for 49 tissues. Our study provides a powerful resource for deciphering the mechanisms of non-coding variation.

## Methods

**The expression modifier score.** Fine-mapping of GTEx v8 data is described in https://www.finucanelab.org/data and is summarized in the Supplementary Methods. We constructed a binary classification task by labeling the variant–gene pairs with PIP > 0.9 for both of the two fine-mapping methods (FINEMAP[19] and Sum of Single Effects, SuSiE[20]) as positive, and the ones with PIP < 0.0001 for both methods as negative. Each variant–gene pair was annotated with 6121 features (distance to TSS annotated in the GTEx v8 dataset, 12 non-cell-type-specific binary features from the LDSC baseline model[6], 795 cell-type-specific binary features from the Roadmap Epigenomics Consortium[5], where variants falling in narrow peak are annotated as 1, and others are 0, and 5313 deep learning-derived cell-type-specific features generated by the Basenji model[28,29]; Supplementary Data 5). The 152 most predictive features were selected based on different prediction accuracy metrics, such as F1 measure and mean decrease of impurity for each feature (Supplementary Methods). A combination of random search followed by grid search was performed to tune the hyperparameter for a random forest classifier that maximizes the AUROC of the binary prediction in the held-out dataset (Supplementary Data 6). Finally, for each prediction score bin, we calculated the fraction of positively labeled samples and scaled the output score, to derive the EMS. Further details are described in the Supplementary Methods.

**Performance evaluation of EMS.** To evaluate the performance of EMS, for each chromosome, we trained EMS using all the other chromosomes to avoid overfitting. CADD[36] v1.4 and GERP[38] scores were annotated using the hail[56] annotation database (https://hail.is), and ncER[39] scores were downloaded from https://github.com/TelentiLab/ncER_datasets. In order to annotate the DeepSEA[26] v1.0 and Fathmm[37] v2.3 noncoding scores, we mapped hg38 coordinates to hg19 using the hail liftover function, removed variants that do not satisfy 1-to-1 matching, and followed their web instructions (https://humanbase.readthedocs.io/en/latest/deepsea.html, and http://fathmm.biocompute.org.uk) to score the variants. Insertion and deletions were not included in the Fathmm scores. For DeepSEA, we calculated the *e*-values from the individual features, following ref. [4]. We computed the area under the receiver operating characteristic curve and the precision recall curve (Supplementary Fig. 5), as well as enrichments of different variant–gene pairs or variants, as described in the next sections (Fig. 3).

**Computation of enrichment.** Enrichment of a specific set of variant–gene pairs (e.g., putative causal variants in GTEx whole blood) in a score bin is defined as the probability of drawing a variant–gene pair in the set given that the variant–gene is in the score bin, divided by the overall probability of drawing a variant–gene pair in the set. The error bar of enrichment denotes the standard error of the numerator, divided by the denominator (we assumed the standard error of the denominator is small enough, since the total number of variant–gene pairs is typically large; >100,000,000 for all the variant–gene pairs in GTEx v8). When testing binary functional features as in Fig. 1, the score is the individual functional feature, and the set is defined by the specific PIP bin.

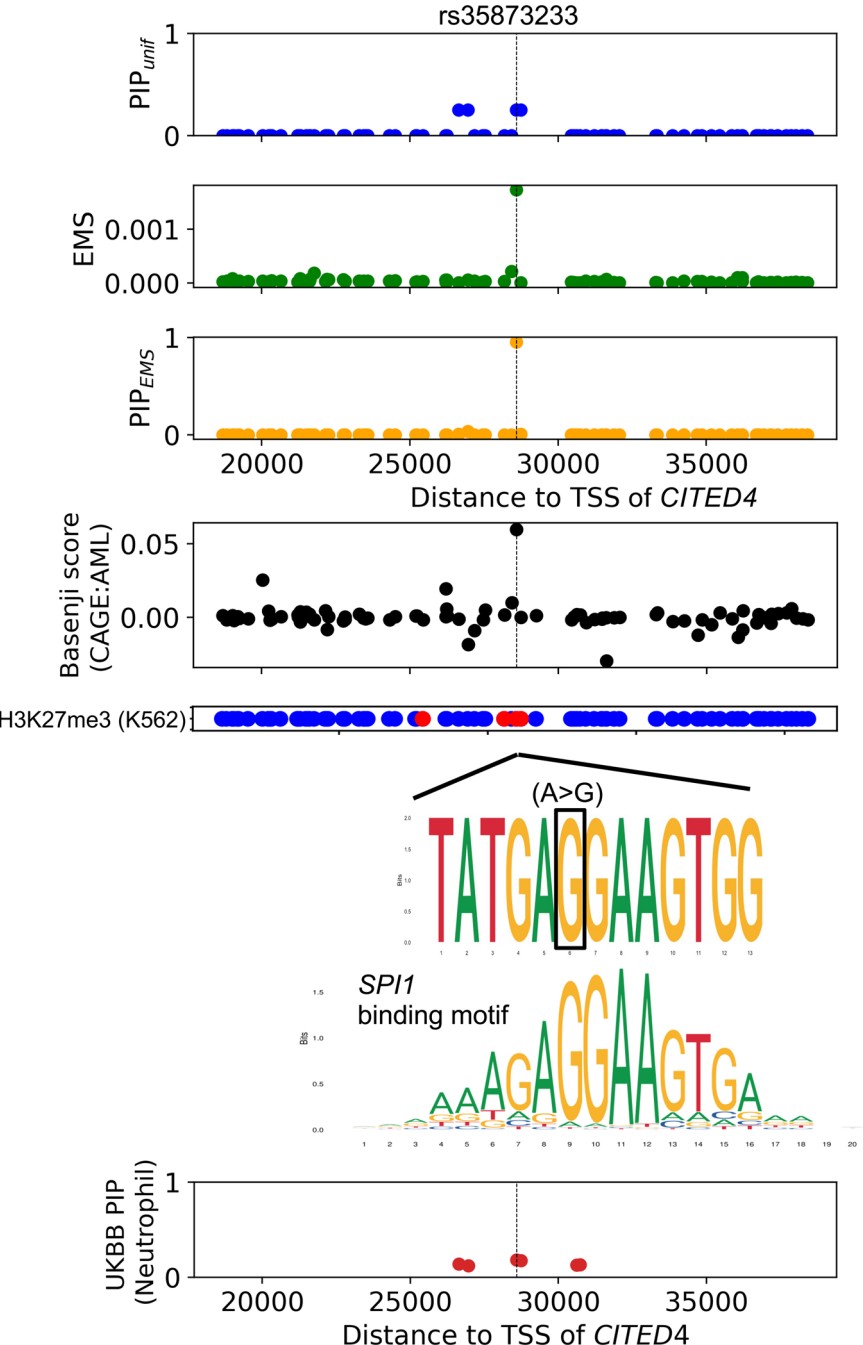

**Fig. 6 An example of a putative causal eQTL prioritized by EMS.** rs35873233, an upstream variant of *CITED4*, was prioritized by functionally informed fine-mapping using EMS as a prior. From top to the bottom: PIP with uniform prior (PIP$_{unif}$), EMS, PIP with EMS as a prior (PIP$_{EMS}$); Basenji score for CAGE[46] activity in acute myeloid leukemia (AML), H3K27me3 narrow peak in K562 cell line (red if the variant is on the peak, blue otherwise), sequence context[60] of the alternative allele aligned with the binding motif[61] of *SPI1*, and PIP for neutrophil count in UKBB (https://www.finucanelab.org/data, ref. [34]) with uniform prior.

**enrichment analysis of eQTL, complex trait, and reporter assay data**.
Saturation mutagenesis data[12] was downloaded from the MPRA data access portal (http://mpra.gs.washington.edu). An MPRA hit was defined as having a bonferroni-significant association *p* value (<0.05 divided by the total number of variant–cell type pairs) for at least one cell type, regardless of the effect size and direction. The raQTL data[40] was downloaded from https://osf.io/w5bzq/wiki/home/. EMS was rescaled to have a constant distance to TSS (200 bp, roughly representing the scale of typical distance to TSS in plasmids[12]), which is expected to significantly decrease the performance of EMS compared to in native genome. Similarly, when comparing EMS with other scores for enrichments of MPRA hits or raQTLs, distance to TSS was not used for the comparison.

Fine-mapping of UKBB traits is described in https://www.finucanelab.org/data. To focus on noncoding regulatory effects, we annotated the variants in VEP[57] v85 and filtered out coding and splice variants for the UKBB dataset. For each (noncoding) variant, we calculated the maximum PIP over all the hematopoietic traits, as well as the maximum whole-blood EMS over all the genes in the *cis*-window of the variant, since a variant can have different regulatory effect on different genes, for different phenotypes. A variant was defined as putative hematopoietic-trait causal if it has SuSiE PIP > 0.9 in any of the hematopoietic traits. In UKBB, we focused on the variants that exist in the GTEx v8 dataset to reduce the calculation complexity.

For all four datasets, the variants (or variant–gene pairs in GTEx) other than putative causal ones were randomly downsampled to achieve a total number of

variants to be exactly 100,000, to reduce the computational burden, while keeping enough number of variants to observe statistical significance. GTEx enrichment, MPRA hits enrichment, raQTL enrichment, and UKBB enrichment are thus defined as the enrichment of putative causal eQTLs, MPRA hits, raQTLs, and putative hematopoietic-trait causal variants in the downsampled dataset, respectively.

**Approximate functionally informed fine-mapping using EMS.** In the SuSiE model, for a given gene, the vector $b$ of true SNP effects on that gene is modeled as a sum of vectors with only one non-zero element each:

$$b = \sum_{l=1}^{L} b_l$$

$$||b_l||_0 = 1$$

where $b$ and $b_l$ are vectors of length $m$ and $m$ is the number of variants in the locus. Intuitively, each $b_l$ corresponds to the contribution of one causal variant. One output of SuSiE is a set of $m$-vectors $\alpha_1, ..., \alpha_L$, with $\alpha_L(v)$ equal to the posterior probability that $b_l(v) \neq 0$; i.e., that the $l$th causal variant is the variant $v$. Credible sets are computed for each $l$ from $\alpha_l$, and credible sets that are not pure—i.e., that contain a pair of variants with absolute correlation < 0.5—are pruned out. The $\alpha_l$ are also used to compute PIPs.

Our algorithm for approximate functionally informed fine-mapping takes the approach of re-weighting the posterior probability calculated using the uniform prior, analogous to ref. [32], and proceeds as follows. For each gene and each tissue, we start with $\alpha_1, ..., \alpha_L$ computed by SuSiE using the uniform prior. For each $l$, if $\alpha_l$ corresponds to a pure credible set, we re-weight each element of $\alpha_l$ by the EMS of the corresponding variant, and we normalize so that the sum is equal to 1, obtaining $\hat{\alpha}_l$. In other words, letting $w_1 ... w_m$ denote the EMSs for the $m$ variants, we define $\hat{\alpha}_l(v)$ for the variant $v$ to be

$$\hat{\alpha}_l(v) = \frac{w_v \alpha_l(v)}{\sum_{u=1}^{m} w_u \alpha_l(u)}$$

if $\alpha_l$ corresponds to a pure credible set; otherwise, we set $\hat{\alpha}_l = \alpha_l$. We then use the updated $\hat{\alpha}_1, ..., \hat{\alpha}_L$ to compute updated PIPs and credible sets, as in the original SuSiE method. See Supplementary Methods for further details.

**Performance evaluation of PIP$_{EMS}$ and application to gene prioritization.** PIP using distance to TSS as a prior (PIP$_{DAP-G}$) was downloaded from the GTEx portal (https://gtexportal.org/). The raQTL data was downloaded from https://osf.io/w5bzq/wiki/home/, and the negative variants were randomly downsampled to a total of 100,000 variants. For complex trait causal noncoding variant prioritization, a threshold of PIP > 0.1 was chosen to account for low sample size. We defined a gene prioritization task using 49 tissues in GTEx v8 and 95 complex traits in UKBB, using the following steps (further details are described in Weeks et al.[43]):

Across all traits, we identified 1 Mb regions centered at unresolved credible sets (no coding variant with PIP > 0.1) that additionally contained at least one "evaluation gene" (protein-coding variant with PIP > 0.5) for the same trait. There were 2897 such regions and 1161 evaluation genes. Our intuition is that the gene with the fine-mapped protein-coding variant is most likely to be the primary causal signal, and that a nearby noncoding signal is more likely to act through this gene (i.e., via regulation) than through a different gene.

For each gene–region pair, we defined the colocalization posterior probability (CLPP) for the gene to be the maximum of the product of the eQTL PIP and trait PIP, across all tissues and all variants in the unresolved credible set. A gene is prioritized if it has CLPP > 0.1 and it has the maximum CLPP in its region. We compute the precision as the number of correctly prioritized genes (where the prioritized gene is also the gene with the primary, protein-coding signal) divided by the total number of prioritized genes. We compute recall as the number of correctly prioritized genes divided by the total number of evaluation genes. The total number of candidate genes is defined as the number of gene–trait pairs, presenting CLPP > 0.1 in at least one tissue and variant.

**Tissue-specific putative causal eQTL analysis.** Tissue-specific putative causal eQTL in a tissue was defined as a variant–gene pair with PIP$_{EMS}$ > 0.9 in the tissue and PIP$_{EMS}$ < 0.1 in all the other tissues (including cases where a variant is missing in a tissue; Supplementary Data 7). A tissue-specific putative causal eQTL pair was defined as a pair of tissue-specific putative causal eQTL on a same gene in two different tissues, existing within 10 kb distance (Supplementary Fig. 14 and Supplementary Data 8). Basenji features were classified as TF related if the feature name contains the gene symbol classified as a human TF in an external database[58] (http://humantfs.ccbr.utoronto.ca/download.php).

Then for each TF, we defined it as specific for tissue $T$ if the expression level (TPM) of the TF was higher in $T$ than in all other tissues and was >2 standard deviations away from the mean expression level across tissues. All the tissues for which the TF had expression level ten times lower than that of tissue $T$ were defined as control tissues. TF-related Basenji features with no specific tissue, or lacking control tissues were filtered out. We also filtered out the features where the TF specificity and the assay cell type did not clearly match (Supplementary Data 9).

This resulted in 42 TF-related Basenji features corresponding to 30 unique TFs. Enrichment of each TF-related Basenji feature was examined by comparing the average score in the tissue-specific putative causal eQTLs for the corresponding tissue with the average in the control tissues, using a $t$ test (Supplementary Data 9).

**Statistical analysis.** All the statistical tests were two-sided. No adjustment was made in the $p$ value we report.

Error bar in Fig. 5b–d and Supplementary Fig. 13 is defined as the standard error of the mean.

Error bar in the enrichment analyses (all the other figures, where error bars are present) are explained in the "Computation of enrichment" section in the "Methods". The set of software used for data generation, statistical analysis, and plotting in the study are listed below:

SuSiE v0.8.1.0521 (https://github.com/stephenslab/susie-paper)
FINEMAP v1.3.1 (http://www.christianbenner.com)
ggseqlogo (https://cran.r-project.org/web/packages/ggseqlogo/index.html)
basenji v0.0.1 (https://github.com/calico/basenji)
brokenaxis v0.3.1 (https://pypi.org/project/brokenaxes/)
joblib v0.11 (https://joblib.readthedocs.io)
hail v0.2.26 (https://hail.is)
matplotlib v3.2.0 (https://matplotlib.org)
numpy v1.18.1 (https://numpy.org)
pandas v1.0.1 (https://pandas.pydata.org)
scikit-learn v0.21.3 and v0.23.2 (https://scikit-learn.github.io/stable)
scipy v1.2.1 (http://scikit-learn.github.io/stable)
seaborn v0.9.0 (https://seaborn.pydata.org).

**Reporting summary.** Further information on research design is available in the Nature Research Reporting Summary linked to this article.

## Data availability

EMS for 49 tissues are available at https://www.finucanelab.org/data. CADD v1.4 and GERP scores were annotated using the hail annotation database (https://hail.is). ncER scores were downloaded from https://github.com/TelentiLab/ncER_datasets. DeepSEA v1.0 scores were downloaded from https://humanbase.readthedocs.io/en/latest/deepsea.html. Fathmm v2.3 noncoding scores were downloaded from http://fathmm.biocompute.org.uk. Saturation mutagenesis data was downloaded from the MPRA data access portal (http://mpra.gs.washington.edu). The raQTL data was downloaded from https://osf.io/w5bzq/wiki/home/. Human transcription factor (TF) data was downloaded from http://humantfs.ccbr.utoronto.ca/download.php. The UKBB fine-mapping results are deposited at https://www.finucanelab.org/data.

## Code availability

Code used in this manuscript is available at https://github.com/FinucaneLab/Expression_Modifier_Score/.

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

## Acknowledgements

We thank Yakir Reshef, Jesse Engreitz, Elle Weeks, and all the members of Finucane lab for useful conversations. H.K.F. was funded by NIH grant DP5 OD024582 and by Eric and Wendy Schmidt. Q.S.W. and M.K. were supported by the Nakajima Foundation Scholarship.

## Author contributions

Q.S.W., D.G.M., and H.K.F. designed the study. Q.S.W., D.R.K., J.U., and S.S. analyzed the data. Q.S.W. and H.K.F. wrote the manuscript with input from all authors (D.R.K., J.U., M.K., S.S., R.C., C.A., N.C., Y.O., B.B.J., F.A., K.G.A., and D.G.M.).

## Competing interests

D.G.M. is a founder with equity in Goldfinch Bio, and has received research support from AbbVie, Astellas, Biogen, BioMarin, Eisai, Merck, Pfizer, and Sanofi-Genzyme.

## Additional information

# The Biobank Japan Project

Koichi Matsuda[12,13], Yuji Yamanashi[14], Yoichi Furukawa[15], Takayuki Morisaki[16], Yoshinori Murakami[17], Yoichiro Kamatani[13,18], Kaori Muto[19], Akiko Nagai[19], Wataru Obara[20], Ken Yamaji[21], Kazuhisa Takahashi[22], Satoshi Asai[23,24], Yasuo Takahashi[25], Takao Suzuki[26], Nobuaki Sinozaki[26], Hiroki Yamaguchi[27], Shiro Minami[28], Shigeo Murayama[29], Kozo Yoshimori[30], Satoshi Nagayama[31], Daisuke Obata[32], Masahiko Higashiyama[33], Akihide Masumoto[34] & Yukihiro Koretsune[35]

[12]Laboratory of Genome Technology, Human Genome Center, Institute of Medical Science, The University of Tokyo, Tokyo, Japan. [13]Laboratory of Clinical Genome Sequencing, Graduate School of Frontier Sciences, The University of Tokyo, Tokyo, Japan. [14]Division of Genetics, The Institute of Medical Science, The University of Tokyo, Tokyo, Japan. [15]Division of Clinical Genome Research, Institute of Medical Science, The University of Tokyo, Tokyo, Japan. [16]Division of Molecular Pathology, IMSUT Hospital Department of Internal Medicine, Institute of Medical Science, The University of Tokyo, Tokyo, Japan. [17]Department of Cancer Biology, Institute of Medical Science, The University of Tokyo, Tokyo, Japan. [18]Laboratory of Complex Trait Genomics, Graduate School of Frontier Sciences, The University of Tokyo, Tokyo, Japan. [19]Department of Public Policy, Institute of Medical Science, The University of Tokyo, Tokyo, Japan. [20]Department of Urology, Iwate Medical University, Iwate, Japan. [21]Department of Internal Medicine and Rheumatology, Juntendo University Graduate School of Medicine, Tokyo, Japan. [22]Department of Respiratory Medicine, Juntendo University Graduate School of Medicine, Tokyo, Japan. [23]Division of Pharmacology, Department of Biomedical Science, Nihon University School of Medicine, Tokyo, Japan. [24]Division of Genomic Epidemiology and Clinical Trials, Clinical Trials Research Center, Nihon University. School of Medicine, Tokyo, Japan. [25]Division of Genomic Epidemiology and Clinical Trials, Clinical Trials Research Center, Nihon University School of Medicine, Tokyo, Japan. [26]Tokushukai Group, Tokyo, Japan. [27]Departmentof Hematology, Nippon Medical School, Tokyo, Japan. [28]Department of Bioregulation, Nippon Medical School, Kawasaki, Japan. [29]Tokyo Metropolitan Geriatric Hospital and Institute of Gerontology, Tokyo, Japan. [30]Fukujuji Hospital, Japan Anti-Tuberculosis Association, Tokyo, Japan. [31]The Cancer Institute Hospital of the Japanese Foundation for Cancer Research, Tokyo, Japan. [32]Center for Clinical Research and Advanced Medicine, Shiga University of Medical Science, Shiga, Japan. [33]Department of General Thoracic Surgery, Osaka International Cancer Institute, Osaka, Japan. [34]IIZUKA-HOSPITAL, Fukuoka, Japan. [35]National Hospital Organization Osaka National Hospital, Osaka, Japan.

