## [Peer Review File · Nature Communications]

Reviewers' Comments:

Reviewer #2:

Remarks to the Author:

The authors present a supervised learning approach to predict putative causal eQTLs. They train random forest models to discriminate a potentially incomplete set of gold standard (putative causal) eQTL variants in several tissues from the GTEx compendium against putative non-causal variants using a large collection of features including tissue specific measures of regulatory activity, distance from TSS of causal gene and variant effect predictions from the Basenji deep learning model. The predictor performs well in held-out chromosome cross-validation experiments. It outperforms previous approaches. They evaluate the importance of different features and derive several interesting insights. Distance to TSS is a very powerful predictor. Basenji variant effect features are collectively stronger but individually weak. Other annotations have lower contributions. EMS performs moderately well in predicting effects of saturated mutagenesis MPRA experiments for several enhancers, reporter assay QTLs and putative causal variants for hematopoietic traits from the UK Biobank. Overall, although EMS is not always the best performing model, the performance is competitive for all these diverse evaluations. Also, the authors are quite clear and honest about the strengths and limitations of their approach in the discussion.

Next, they calibrate the predictor and use the predicted EMS probabilities within a functionally informed fine mapping approach to fine map GTEx eQTLs across 49 tissues. They obtain 20,913 additional QTLs with high posterior probabilities. The authors don't present any direct validation of any of these new predictions. However, they do show that these EMS prior supported fine mapped variants do improve GWAS colocalization and gene prioritization using gold standard gene sets for several UKBB phenotypes, relative to a fine mapped variants using uniform priors. However, TSS based priors were competitive with EMS based priors. I think this section is a bit weak and some amount of functional validation even at the level of effects on predicted target gene expression of the novel additional fine mapped variants would potentially have provided more confidence in the results. The CITED4 vignette the authors is great. In the absence of experimental validation, this kind of supporting evidence for the other predicted fine mapped variants would have been really nice.

- Are there eQTLs shared across tissues that fine map to different variants supported by different tissue-specific regulatory features?

- The Basenji model can be interpreted to figure out which sequence features are likely disrupted by candidate fine mapped variants. This would have been a fantastic addition to provide support for their variants. One would expect classic tissue-specific TF binding sites being disrupted for tissue-specific causal eQTLs.

Overall, I think this method is clever and provides a promising direction for future efforts. But as stated above, I feel like there is a missed opportunity to provide more mechanistic hypotheses about these fine mapped variants and candidate causal genes which would have strengthened the paper in the absence of actual experimental validation.

However, I am sufficiently convinced that the results are robust and of sufficiently high value. Hence, I leave it to the authors to decide whether they would like to put in the additional effort to provide supporting evidence for their novel variants and genes.

Reviewer #3:

Remarks to the Author:

The authors developed a novel machine learning approach using statistically fine-mapped variants to train supervised models for causal QTL variant prioritization. Using this approach and functional variant features, the authors compute a variant-specific score called EMS, representing the probability of variant being causal. EMS showed good performance in cross-validation and enrichment of reporter

assay QTLs and putative causal variants in GWAS. The authors further applied EMS to compute an adjusted fine-mapping posterior inclusion probability score, named PIP-EMS. PIP-EMS identifies more likely causal variants when jointly analyzed with GWAS data than applying PIP with a uniform prior. Overall, this manuscript described a novel approach and this work is well designed, properly evaluated and well written. I only have minor suggestions for the manuscript.

1. Feature importance of EMS. Even though the post hoc feature importance analysis provides a useful reference, it can be more informative to compare EMS performance trained using different subsets of features (e.g. TSS + binary annotations).

2. Supp File 5 - Roadmap tab seems to be missing features (only H3K9me3 was listed).

**Leveraging supervised learning for functionally-informed fine-mapping of cis-eQTLs
identifies an additional 20,913 putative causal eQTLs
(Wang et al, NCOMMS-20-42676A)
Response to reviewers**

We thank the reviewers for their thoughtful comments and have made a number of changes to the manuscript in response, which have substantially improved this work.

Response to Reviewer #2:

We appreciate the reviewer's detailed summary of our work, positive evaluation and thoughtful questions.

I feel like there is a missed opportunity to provide more mechanistic hypotheses about these fine mapped variants and candidate causal genes which would have strengthened the paper in the absence of actual experimental validation.

- Are there eQTLs shared across tissues that fine map to different variants supported by different tissue-specific regulatory features?

- The Basenji model can be interpreted to figure out which sequence features are likely disrupted by candidate fine mapped variants. This would have been a fantastic addition to provide support for their variants. One would expect classic tissue-specific TF binding sites being disrupted for tissue-specific causal eQTLs.

We appreciate the reviewer's point. We have addressed these two questions below and in our revised manuscript, and we believe our manuscript is stronger as a result. We will begin with the second question and then address the first question.

To address the second point, we first defined a set of tissue-specific TFs from the expression data in GTEx v8. We then identified 42 Basenji features corresponding to the activity of these tissue-specific TFs measured in a matched cell type. For example, we identified CEBPZ as LCL-enriched in GTEx v8, and so we included a Basenji feature corresponding to CEBPZ activity in GM12878 but not CEBPZ activity in HepG2. All 42 features corresponded to one of three tissues (LCL, Liver and Whole Blood). For each of these 42 features, we computed the average feature value in tissue-specific putative causal eQTLs in the matched tissue as well as the average feature value in tissue-specific putative causal eQTLs in control tissues. As the reviewer expected, this analysis clearly highlighted the enrichment of tissue-specific TF activity in the corresponding set of tissue-specific putative causal eQTLs. We have added this result as **Fig. 5b-d** and in the main text on page 4: *"Tissue-specific putative causal eQTLs were characterized by enrichments of corresponding tissue-specific transcription factor activity scores in the Basenji model (Figs. 5b-d, S13, S14; Methods)."*

b-d. Mean Basenji score in different classes of tissue-specific putative causal eQTLs for tissue-specific TF-related Basenji features for liver (**b**), whole blood (**c**), and LCLs (**d**). In 39 out of all 42 features across all three tissues, the mean Basenji score in tissue-specific putative causal eQTLs identified by PIP_{EMS} is significantly higher in the corresponding tissue than in control tissues (t-test $p < 0.05/42$). This changes to 36 in 42 when using PIP_{unif} instead of PIP_{EMS}. The enrichment of mean Basenji score in putative causal eQTLs in the corresponding tissue compared to control tissues is higher for PIP_{EMS} than PIP_{unif} for all 42 tissues ($p < 10^{-100}$ in aggregate), consistent with our understanding that functionally-informed fine-mapping using EMS utilizes cell type specific functional enrichments, identified from putative causal eQTLs identified with a uniform prior, to identify additional putative causal eQTLs. Duplicated names are distinct features corresponding to biological replicates in the TF activity measurements.

In this figure, we also plotted the Basenji score enrichments in tissue-specific putative causal eQTLs identified by statistical fine-mapping with uniform prior, as a comparison. The comparison shows that Basenji score enrichments are also present in putative causal eQTLs identified with uniform prior, and that those identified with EMS as a prior show a higher enrichment of TF scores than those identified with a uniform prior. This is consistent with our understanding that functionally-informed fine-mapping using EMS utilizes cell type specific functional enrichments, identified from putative causal eQTLs identified with a uniform prior, to identify additional putative causal eQTLs. We added such descriptions in the figure legend above.

The skewed distribution of tissue-specific TF-related Basenji features in terms of the variety of TF specificity and the assay cell types (as shown in the figure below, not in the manuscript) did not allow us to quantify such enrichments for other tissues. For example, there are 14 features corresponding to neural cell types, including SK-N-SH and SK-N-MC, but they do not contain TFs specific to brain-related tissues. The 42 features included in our analysis were the only ones available for which the TF was tissue specific and measured in a matching cell type.

Figures: Distribution of the matched tissue of tissue-specific TFs (a) or cell types of the corresponding CHIP assays (b) in 454 TF-related Basenji features. Tissues and cell types with zero features are not shown.

To highlight the importance of quantifying TF scores in the matched cell type, we also added a figure panel showing that the enrichment is limited when the cell type does not match the specificity of the TF as **Fig. S13**:

Fig. S13. Comparing Basenji scores characterized by same TFs in different cell types
 Comparing the average LCL-specific TF-related Basenji score in LCL-specific putative causal eQTLs when the TF was assayed in LCLs (circle dots) or in a non-LCL cell type (triangle dots). The score enrichment is minimal when the cell type does not match the TF specificity (e.g. MCF-7, a breast cancer cell line), even when the tissue specificity of the TF and the tissue specificity of the eQTL match (e.g., E2F8 and LCLs).

We also made the full list of tissue-specific putative causal eQTLs available as **Supplementary File 7 (List of tissue-specific putative causal eQTLs)** for the readers interested in pursuing further functional validation in other tissues.

Next, in order to address the first point, we further explored the full list of tissue-specific putative causal eQTLs, to identify pairs of nearby (<10 kb) putative causal eQTLs of a single gene that are tissue-specific in distinct tissues (n= 3,102 pairs). We made the list available as **Supplementary File 8 (List of pairs of tissue-specific putative causal eQTLs for the same gene)**. We then investigated individual loci to understand the underlying tissue-specific regulatory features' contribution, and highlighted examples of two loci that are well characterized by tissue-specific TF binding activity differences as **Fig. S14**:

Fig S14. Examples of pairs of nearby putative causal eQTLs for a single gene that are tissue-specific in distinct tissues.

In **a**, the Whole Blood-specific putative causal eQTL is characterized by a Basenji score corresponding to *SPI1* activity. In **b**, the Whole Blood or Liver-specific putative causal eQTLs are characterized by a Basenji score corresponding to *CEBPB* or *HNF4A* activity, respectively. (Negative Basenji score corresponds to disruption.) Each of the causal tissues matches the specificity of the corresponding TF.

A step-by-step description of these analyses was added in the method section (“Tissue-specific putative causal eQTL analysis”, page 17-18):

Tissue-specific putative causal eQTL analysis

A tissue-specific putative causal eQTL in a tissue was defined as a variant-gene pair with $PIP_{EMS} > 0.9$ in the tissue and $PIP_{EMS} < 0.1$ in all the other tissues (including cases where a variant is missing in a tissue) (**Supplementary File 7**). A tissue-specific putative causal eQTL pair was defined as a pair of tissue-specific putative causal eQTLs for the same gene in two different tissues, within 10 kb distance (**Fig. S14, Supplementary File 8**). Basenji features were classified as transcription factor (TF) related if the feature name contains the gene symbol classified as a human transcription factor in an external database⁵⁹ (<http://humantfs.ccb.utoronto.ca/download.php>).

Then for each TF, we defined it as specific for tissue *T* if the expression level (TPM) of the TF was higher in *T* than in all other tissues and was more than 2 standard deviations away from the mean expression level across tissues. All the tissues for which the TF had expression level 10 times lower than that of tissue *T* were defined as control tissues. TF-related Basenji features with no specific tissue, or lacking control tissues were filtered out. We also filtered out the

features where the TF specificity and the assay cell type did not clearly match (**Supplementary File 9**). This resulted in 42 TF-related Basenji features corresponding to 30 unique TFs. Enrichment of each TF-related Basenji feature was examined by comparing the average score in the tissue-specific putative causal eQTLs for the corresponding tissue with the average in the control tissues using a t-test (**Supplementary File 9**).

In conclusion, we investigated tissue-specific putative causal eQTLs both collectively and individually to show that they can be characterized by tissue-specific TF-related features. We again would like to thank the reviewer for providing excellent suggestions.

Response to Reviewer #3:

We appreciate the reviewer's concise summary of our work, positive evaluation and valuable suggestions.

1. Feature importance of EMS. Even though the post hoc feature importance analysis provides a useful reference, it can be more informative to compare EMS performance trained using different subsets of features (e.g. TSS + binary annotations).

This is a great suggestion. We have expanded the **a** and **b** panel of **Fig. S20** (now **Fig. S22**) to include the analysis suggested by the reviewer.

The result here is consistent with our feature importance analysis, in that distance to TSS is the most important feature (and thus most predictive single feature), followed by Basenji features. Since Basenji features are correlated by nature, using Basenji features alone does not result in predictive power as high as using distance to TSS alone. Our results also show that the effect of excluding binary features is minimal.

2. Supp File 5 - Roadmap tab seems to be missing features (only H3K9me3 was listed).

Thank you so much for reviewing our supplementary files in detail. There was a careless mistake indeed, and this has been corrected. We have updated Supplementary File 5.

We thank again for the reviewer's valuable suggestions that improved the manuscript.

Reviewers' Comments:

Reviewer #2:

Remarks to the Author:

Thanks to the authors for putting in the extra effort to address my comments. I am satisfied by the response. Great paper. Very useful resource.

Signed

-Anshul Kundaje

Reviewer #3:

Remarks to the Author:

I thank the authors for the revision. My points have been addressed.

**Leveraging supervised learning for functionally-informed fine-mapping of cis-eQTLs
identifies an additional 20,913 putative causal eQTLs
(Wang et al, NCOMMS-20-42676A)
Final response to reviewers
2021/04/02**

We thank the reviewers for their thoughtful comments that enabled us to substantially improve the manuscript:

Reviewer #2 (Remarks to the Author):

Thanks to the authors for putting in the extra effort to address my comments. I am satisfied by the response. Great paper. Very useful resource.

***Signed
-Anshul Kundaje***

The comments were extremely helpful, and we are happy to hear your positive response. Thank you so much again for reviewing our manuscript and providing excellent insights.

Reviewer #3 (Remarks to the Author):

I thank the authors for the revision. My points have been addressed.

Thank you so much again for reviewing our manuscript in detail and raising valuable points.